# Quantum Incoherence Based Simultaneously on *k* Bases

**DOI:** 10.3390/e24050659

**Published:** 2022-05-07

**Authors:** Pu Wang, Zhihua Guo, Huaixin Cao

**Affiliations:** School of Mathematics and Statistics, Shaanxi Normal University, Xi’an 710119, China; wangpu@snnu.edu.cn

**Keywords:** strong incoherence, weak coherence, orthonormal basis, mutually unbiased basis

## Abstract

Quantum coherence is known as an important resource in many quantum information tasks, which is a basis-dependent property of quantum states. In this paper, we discuss quantum incoherence based simultaneously on *k* bases using Matrix Theory Method. First, by defining a correlation function m(e,f) of two orthonormal bases *e* and *f*, we investigate the relationships between sets I(e) and I(f) of incoherent states with respect to *e* and *f*. We prove that I(e)=I(f) if and only if the rank-one projective measurements generated by *e* and *f* are identical. We give a necessary and sufficient condition for the intersection I(e)⋂I(f) to include a state except the maximally mixed state. Especially, if two bases *e* and *f* are mutually unbiased, then the intersection has only the maximally mixed state. Secondly, we introduce the concepts of strong incoherence and weak coherence of a quantum state with respect to a set B of *k* bases and propose a measure for the weak coherence. In the two-qubit system, we prove that there exists a maximally coherent state with respect to B when k=2 and it is not the case for k=3.

## 1. Introduction

Quantum coherence is not only a feature of quantum systems which arise due to superposition principle, but also is a kind of fundamental resources in quantum information and computation [1,2,3,4,5,6,7,8]. The resource theory of coherence is formulated with respect to a distinguished basis of a Hilbert space, which defines free states as the states that are diagonal in this basis [3]. Several important quantifiers of quantum coherence have been introduced and assessed [9,10,11,12,13,14,15,16,17,18,19]. Recently, it is shown that quantum coherence can be useful resource in quantum computation [20,21,22,23,24], quantum metrology [25], quantum thermodynamics [26,27,28,29,30,31] and quantum biology [32,33,34].

Since the coherence of quantum states depends on the choice of the reference basis, it is natural to study the relationship among the coherence with respect to different bases. Cheng et al. [35] first studied the situation of two specific coherence measures under mutual unbiased basis (MUB): $\ell_{1}$ norm of coherence and relative entropy of coherence. They proposed the complementary relationship of the two coherence measures under any complete MUB set. Rastegin in [36] discussed the uncertainty relation for the geometric measure of coherence under MUBs. Sheng et al. [37] further studied the realization of quantum coherence through skewed information and the geometric measure under mutual unbiased bases. Recently, considered the standard coherence (SC), the partial coherence (PC) [38,39,40] and the block coherence (BC) [41,42] as variance of quantum states under some quantum channel Φ, Zhang et al. [43] proposed the concept of channel-based coherence of quantum states, called Φ-coherence, which contains the SC, PC and BC, but not contain the POVM-based coherence [44,45], and obtained some interesting results.

Usually, the coherence of an individual quantum state is discussed only when referring to a preferred basis. Considered sets of quantum states, Designolle et al. [46] introduced the concept of set coherence for characterizing the coherence of a set of quantum states in a basis-independent way. Followed a resource-theoretic approach, the authors of [46] defined the free sets of states as sets $F_{n}$ of groups of states $\vec{\rho }={\left\{\rho_{j}\right\}}_{j=1}^{n}$ such that there exists a choice of basis (equivalently, a unitary *U*) for which all states $U\rho_{j}U^{†}$ in the set $U\vec{\rho }U^{†}$ become diagonal. Clearly, $\vec{\rho }\in F_{n}$ if and only if ${\left\{\rho_{j}\right\}}_{j=1}^{n}$ is a commutative family of states, i.e., $\rho_{i}\rho_{j}=\rho_{j}\rho_{i}$ for all i,j=1,2,…,n.

Different from the discussions above, in this paper, we focus on the quantum incoherence based simultaneously on *k* bases; equivalently, the coherence of a quantum state with respect to a basis contained in a given set B of *k* orthonormal bases. In Section 2, by defining the correlation function of two orthonormal bases *e* and *f*, we study the relationships between two sets of incoherent states with respect to *e* and *f*, and investigate the maximally coherent states with respect to *e* and *f*. In Section 3, we discuss the strong incoherence and the weak coherence of a state with respect to a set of *k* orthonormal bases and introduce a measure for the weak coherence. In Section 4, we give a summary of this paper.

## 2. Correlation Function of Two Bases and Quantum Coherence

Let us consider a quantum system *X*, which is described by a *d*-dimensional Hilbert space *H* and let *I* denote the identity operator on *H*. We use B(H) and D(H) to denote the sets of all linear operators and all density operators (mixed states) on *H*, respectively. In quantum information theory, a positive operator valued measure (POVM) is a set $M={\left\{M_{i}\right\}}_{i=1}^{m}$ of operators on *H* with $0\leq M_{i}\leq I$ for all i=1,2,…,m and $\sum_{i=1}^{m}M_{i}=I.$ In particular, if $M_{i}^{2}=M_{i}$ for all *i*, then the POVM becomes a projective measurement (PM). For a rank-one PM *P*, there exists an orthonormal basis $e=\{|e_{i}\rangle {\}}_{i=1}^{d}$ such that $P=\{|e_{i}\rangle {\langle e_{i}|\}}_{i=1}^{d}$. In this case, we denote $P=P_{e}=\{|e_{i}\rangle {\langle e_{i}|\}}_{i=1}^{d}$. We use the notation $\bar{z}$ or $z^{*}$ to denote the conjugate of a complex number *z*.

For the fixed orthonormal basis $e=\{|e_{i}\rangle {\}}_{i=1}^{d}$ for *H*, I(e) denotes the set of incoherent states on *H* w.r.t. *e*, i.e., ones that have diagonal matrix representation under the basis *e*. A quantum operation Φ on B(H) is said to be an incoherent operation [3] w.r.t *e* if it admits an incoherent Kraus decomposition, i.e.,
$$\Phi \left(\rho \right)=\sum_{i=1}^{n}K_{i}\rho K_{i}^{†},\;\;\forall \rho \in B\left(H\right)$$
with
$$K_{i}\rho K_{i}^{†}\in tr\left(K_{i}\rho K_{i}^{†}\right)I\left(e\right),\;\;\forall \rho \in I\left(e\right),i=1,2,\ldots ,n.$$

We use IO(e) to denote the set of incoherent operations w.r.t *e* on B(H).

According to Ref. [3], a coherence measure with respect to *e*, called an *e*-coherence measure, is a function C:D(H)↦R satisfying the following four conditions.

(1) Faithfulness: C(ρ)≥0 for all ρ∈D(H); C(ρ)=0 if and only if ρ∈I(e).

(2) Monotonicity: C(Φ(ρ))≤C(ρ) for any Φ∈IO(e).

(3) Strong monotonicity: ∀ρ∈D(H), $\sum_{i=1}^{n}p_{i}C\left(\rho_{i}\right)\leq C\left(\rho \right)$ for all operators $K_{i}$ in H such that $\sum_{i=1}^{n}K_{i}^{†}K_{i}=I$ with $K_{i}I\left(e\right)K_{i}^{†}\subset R^{+}I\left(e\right)$, $p_{i}=\mathrm{tr}\left(K_{i}\rho K_{i}^{†}\right)$ and $\rho_{i}=K_{i}\rho K_{i}^{†}/p_{i}$ if $p_{i}>0$; $\rho_{i}=\frac{1}{d}I$ if $p_{i}=0$.

(4) Convexity: $C\left(\sum_{i=1}^{n}p_{i}\rho_{i}\right))\leq \sum_{i=1}^{n}p_{i}C\left(\rho_{i}\right)$ for any states $\rho_{i}\in D\left(H\right)\left(i=1,2,\ldots ,n\right)$ and any probability distribution ${\left\{p_{i}\right\}}_{i=1}^{n}$.

A usual $\ell_{1}$-norm coherence measure [3] of a state ρ∈D(H) with respect to a basis *e* is defined by
$$C_{e,\ell_{1}}\left(\rho \right)=2\sum_{1\leq i<j\leq n}|\langle e_{i}\left|\rho \right|e_{j}\rangle |.$$
Clearly,
(1)$$C_{e,\ell_{1}}\left(\rho \right)=\sum_{i,j=1}^{n}|\langle e_{i}\left|\rho \right|e_{j}\rangle |-1\leq d-1.$$
Especially, $C_{e,\ell_{1}}\left(\rho \right)=d-1$ if and only if $|\langle e_{i}\left|\rho \right|e_{j}\rangle |=\frac{1}{d}$ for all i,j=1,2,…,d; in that case, ρ is called a *maximally coherent state* with respect to *e*.

From the review above, we find that quantum coherence relies on the choice of orthonormal bases. In what follows, we discuss the relationship between quantum coherence based on different reference bases. To do this, we let $e=\{|e_{i}\rangle {\}}_{i=1}^{d}$ and $f=\{|f_{j}\rangle {\}}_{i=1}^{d}$ be two orthonormal bases for *H* and define
(2)$$m\left(e,f\right)=\sum_{i,j=1}^{d}\left|\langle e_{i}|f_{j}\rangle \right|-d,$$
called the *correlation function* between two bases *e* and *f*.

Recall that [35] two orthonormal bases *e* and *f* for *H* are said to be mutually unbiased if $|\langle e_{i}|f_{j}\rangle |=\frac{1}{\sqrt{d}}$ for all i,j=1,2,…,d. Thus, when *e* and *f* for *H* are mutually unbiased, it holds that $m\left(e,f\right)=d^{\frac{3}{2}}-d$. More properties of the correlation function are given in the following theorem.

**Theorem** **1.**
*Let e and f be two orthonormal bases for H. Then*

*(1) $0\leq m\left(e,f\right)\leq d^{\frac{3}{2}}-d.$*

*(2) m(e,f)=0 if and only if $P_{e}=P_{f}$ if and only if I(e)=I(f).*

*(3) $m\left(e,f\right)=d^{\frac{3}{2}}-d$ if and only if e and f are mutually unbiased bases.*


**Proof.** (1) Since $0\leq |\langle e_{i}|f_{j}\rangle |\leq 1,$ we get $|\langle e_{i}|f_{j}\rangle {|}^{2}\leq \left|\langle e_{i}|f_{j}\rangle \right|$ for all i,j=1,2,…,d. So,
$$\begin{array}{cc} \sum_{i,j=1}^{d}\left|\langle e_{i}|f_{j}\rangle \right| & \geq \sum_{i,j=1}^{d}{\left|\langle e_{i}|f_{j}\rangle \right|}^{2}\\ \; & =\sum_{j=1}^{d}\left(\sum_{i=1}^{d}{\left|\langle e_{i}|f_{j}\rangle \right|}^{2}\right)\\ \; & =\sum_{j=1}^{d}\left\|\right|f_{j}{\rangle \|}^{2}\\ \; & =d. \end{array}$$
This shows that m(e,f)≥0. Since $e=\{|e_{i}\rangle {\}}_{i=1}^{d}$ and $f=\{|f_{j}\rangle {\}}_{i=1}^{d}$ are two orthonormal bases for *H*, there exists a d×d unitary matrix $U=[\lambda_{ij}]$ such that $(|e_{1}\rangle ,|e_{2}\rangle ,\ldots ,|e_{d}\rangle )=U(|f_{1}\rangle ,|f_{2}\rangle ,\ldots ,|f_{d}\rangle );$ equivalently,
(3)$$|e_{i}\rangle =\sum_{j=1}^{d}\lambda_{ij}|f_{j}\rangle ,\;\forall i=1,2,\ldots ,d.$$
Hence, $\lambda_{ij}=\langle f_{j}|e_{i}\rangle$, and using the Cauchy inequality yields that
$$\begin{array}{cc} \sum_{i,j=1}^{d}\left|\langle e_{i}|f_{j}\rangle \right| & =\sum_{i,j=1}^{d}\left|\lambda_{ij}\right|\\ \; & =\sum_{i=1}^{d}\left(\sum_{j=1}^{d}1\cdot \left|\lambda_{ij}\right|\right)\\ \; & \leq \sum_{i=1}^{d}\sqrt{d}\sqrt{\sum_{j=1}^{d}{\left|\lambda_{ij}\right|}^{2}}\\ \; & =d^{\frac{3}{2}}. \end{array}$$
Consequently, $m\left(e,f\right)\leq d^{\frac{3}{2}}-d.$(2) We see from Equation (Equation 2) that m(e,f)=0 if and only if for any *i*, there exists a unique $i^{\prime }$ such that $|\langle e_{i}|f_{i^{\prime }}\rangle |=1$ and $|\langle e_{i}|f_{k}\rangle |=0$ for all $k\neq i^{\prime }$ if and only if for any *i*, there exists a unique $i^{\prime }$ such that $|e_{i}\rangle =e^{i\theta_{ii^{\prime }}}|f_{i^{\prime }}\rangle$, which is equivalent to $P_{e}=P_{f}$, i.e., I(e)=I(f).(3) From the proof of (1), we see that $m\left(e,f\right)=d^{\frac{3}{2}}-d$ if and only if $|\lambda_{ij}|=\frac{1}{\sqrt{d}}\left(\forall i,j\right)$, that is, *e* and *f* are mutually unbiased bases.Suppose that *e* and *f* are mutually unbiased bases, then the coefficients $\lambda_{ij}$ in (Equation 3) satisfy $|\lambda_{ij}\left|=\right|\langle f_{j}|e_{i}\rangle |=\frac{1}{\sqrt{d}}$ for all i,j=1,2,…,d. Let ρ∈I(e)∩I(f). Then it can be written as $\rho =\sum_{n=1}^{d}\mu_{n}|e_{n}\rangle \langle e_{n}|$ with $\mu_{n}\geq 0$ for all n=1,2,…,d, $\sum_{n=1}^{d}\mu_{n}=1$. Using Equation (Equation 3) implies that
$$\rho =\sum_{j,k=1}^{d}\sum_{n=1}^{d}\mu_{n}\bar{\lambda_{nj}}\lambda_{nk}|f_{j}\rangle \langle f_{k}|.$$
Since ρ∈I(f) and $\sum_{n=1}^{d}\mu_{n}=1$, we see that
$$\sum_{n=1}^{d}\mu_{n}\bar{\lambda_{nj}}\lambda_{nk}=\frac{1}{d}\delta_{k,j},\;\;\forall k,j=1,2,\ldots ,d$$
that is,
$$\left(\begin{array}{cccc} \bar{\lambda_{11}} & \bar{\lambda_{21}} & \cdots & \bar{\lambda_{d1}}\\ \bar{\lambda_{12}} & \bar{\lambda_{22}} & \cdots & \bar{\lambda_{d2}}\\ \vdots & \vdots & \ddots & \vdots \\ \bar{\lambda_{1d}} & \bar{\lambda_{2d}} & \cdots & \bar{\lambda_{dd}} \end{array}\right)\left(\begin{array}{cccc} \mu_{1} & 0 & 0 & 0\\ 0 & \mu_{2} & 0 & 0\\ \vdots & \vdots & \ddots & \vdots \\ 0 & 0 & 0 & \mu_{d} \end{array}\right)\left(\begin{array}{cccc} \lambda_{11} & \lambda_{12} & \cdots & \lambda_{1d}\\ \lambda_{21} & \lambda_{22} & \cdots & \lambda_{2d}\\ \vdots & \vdots & \ddots & \vdots \\ \lambda_{d1} & \lambda_{d2} & \cdots & \lambda_{dd} \end{array}\right)=\left(\begin{array}{cccc} \frac{1}{d} & 0 & 0 & 0\\ 0 & \frac{1}{d} & 0 & 0\\ \vdots & \vdots & \ddots & \vdots \\ 0 & 0 & 0 & \frac{1}{d} \end{array}\right).$$
Since $U=[\lambda_{ij}]$ is a d×d unitary matrix, we get $\mu_{k}=\frac{1}{d}$ for all k=1,2,…,d, i.e., $\rho =\frac{1}{d}\sum_{j=1}^{d}|f_{j}\rangle \langle f_{j}|=\frac{1}{d}I$. Hence, $I\left(e\right)\cap I\left(f\right)=\left\{\frac{1}{d}I\right\}$.□

**Remark** **1.**
*Suppose that $P_{e}\neq P_{f}$, then there exists an i and $j_{1},j_{2},\ldots ,j_{k}\left(2\leq k\leq d\right)$ such that $\langle e_{i}|f_{j_{s}}\rangle \neq 0\left(s=1,2,\ldots ,k\right)$ and*

$$|e_{i}\rangle =\sum_{s=1}^{k}\langle f_{j_{s}}|e_{i}\rangle |f_{j_{s}}\rangle .$$

*Then $|e_{i}\rangle \langle e_{i}|\in I\left(e\right)$ and*

$$|e_{i}\rangle \langle e_{i}|=\sum_{s=1,t=1}^{k}\langle f_{j_{s}}|e_{i}\rangle \bar{\langle f_{j_{t}}|e_{i}\rangle }|f_{j_{s}}\rangle \langle f_{j_{t}}|.$$

*Since $\langle f_{j_{s}}|e_{i}\rangle \bar{\langle f_{j_{t}}|e_{i}\rangle }\neq 0$ for any s≠t, we get that $|e_{i}\rangle \langle e_{i}|\notin I\left(f\right).$ This shows that there exists a state ρ∈I(e) but ρ∉I(f). Similarly, there also exists a state $\rho^{\prime }\in I\left(f\right)$ but $\rho^{\prime }\notin I\left(e\right)$.*


From Theorem 1 and Remark 1, we get relationships between m(e,f) and I(e)⋂I(f) as shown by the following Figure 1.

It is clear that $\frac{1}{d}I\in I\left(e\right)⋂I\left(f\right)$ for any bases *e* and *f*. Especially, $I\left(e\right)⋂I\left(f\right)=\left\{\frac{1}{d}I\right\}$ if they are mutually unbiased. However, even though *e* and *f* are not a pair of mutually unbiased bases, it is possible that $I\left(e\right)⋂I\left(f\right)=\left\{\frac{1}{d}I\right\},$ see the following example.

**Example** **1.**
*Let e={|0〉,|1〉} and $f=\left\{|f_{0}\rangle ,|f_{1}\rangle \right\}$ be two orthonormal bases for $H=C^{2}$ with*

$$|f_{0}\rangle =\frac{1}{\sqrt{3}}|0\rangle +\frac{\sqrt{2}}{\sqrt{3}}\left|1\rangle ,\right|f_{1}\rangle =-\frac{\sqrt{2}}{\sqrt{3}}|0\rangle +\frac{1}{\sqrt{3}}|1\rangle .$$

*Clearly, e and f are not a pair of mutually unbiased bases while $I\left(e\right)⋂I\left(f\right)=\left\{\frac{1}{2}I\right\}$.*


This example leads us to study the relationship between two bases *e* and *f* for *H* such that
$$I\left(e\right)⋂I\left(f\right)=\left\{\frac{1}{d}I\right\}.$$
To do this, we let $e=\{|e_{i}\rangle {\}}_{i=1}^{d}$ and $f=\{|f_{i}\rangle {\}}_{i=1}^{d}$ be two bases for *H* and $\rho =\sum_{i=1}^{d}x_{i}|e_{i}\rangle \langle e_{i}|\in I\left(e\right)\backslash \left\{I/d\right\}.$ Since $x_{1},\ldots ,x_{d}$ are the eigenvalues of ρ, they can be rearranged as $\lambda_{1},\lambda_{2},\ldots ,\lambda_{d}$ in decreasing order, say, $\lambda_{1}\geq \lambda_{2}\geq \ldots \geq \lambda_{d}$. Thus, there exists a permutation matrix $P_{1}$ such that
(4)$$P_{1}\left(\begin{array}{c} x_{1}\\ x_{2}\\ \vdots \\ x_{d} \end{array}\right)=\left(\begin{array}{c} \lambda_{1}\\ \lambda_{2}\\ \vdots \\ \lambda_{d} \end{array}\right).$$
Suppose that ρ∈I(f). Then
(5)$$\rho =\sum_{j=1}^{d}y_{j}|f_{j}\rangle \langle f_{j}|,$$
where $y_{j}=\langle f_{j}\left|\rho \right|f_{j}\rangle$. Using Equation (Equation 5) implies that
$$x_{i}=\langle e_{i}\left|\rho \right|e_{i}\rangle =\sum_{j=1}^{d}{\left|\langle e_{i}|f_{j}\rangle \right|}^{2}y_{j}\left(i=1,2,\ldots ,d\right),$$
i.e.,
(6)$$\left(\begin{array}{c} x_{1}\\ x_{2}\\ \vdots \\ x_{d} \end{array}\right)=C\left(\begin{array}{c} y_{1}\\ y_{2}\\ \vdots \\ y_{d} \end{array}\right),$$
where
(7)$$C=\left(\begin{array}{cccc} |\langle e_{1}|f_{1}\rangle {|}^{2} & |\langle e_{1}|f_{2}\rangle {|}^{2} & \cdots & |\langle e_{1}|f_{d}\rangle {|}^{2}\\ |\langle e_{2}|f_{1}\rangle {|}^{2} & |\langle e_{2}|f_{2}\rangle {|}^{2} & \cdots & |\langle e_{2}|f_{d}\rangle {|}^{2}\\ \vdots & \vdots & \ddots & \vdots \\ |\langle e_{d}|f_{1}\rangle {|}^{2} & |\langle e_{d}|f_{2}\rangle {|}^{2} & \cdots & |\langle e_{d}|f_{d}\rangle {|}^{2} \end{array}\right).$$
Since $y_{1},\ldots ,y_{d}$ are also the eigenvalues of ρ, they can be also rearranged as $\lambda_{1},\lambda_{2},\ldots ,\lambda_{d}$ in decreasing order. So, there exists a permutation matrix $P_{2}$ such that
(8)$$P_{2}\left(\begin{array}{c} y_{1}\\ y_{2}\\ \vdots \\ y_{d} \end{array}\right)=\left(\begin{array}{c} \lambda_{1}\\ \lambda_{2}\\ \vdots \\ \lambda_{d} \end{array}\right).$$
Thus,
(9)$$\left(\begin{array}{c} \lambda_{1}\\ \lambda_{2}\\ \vdots \\ \lambda_{d} \end{array}\right)=P_{1}\left(\begin{array}{c} x_{1}\\ x_{2}\\ \vdots \\ x_{d} \end{array}\right)=P_{1}C\left(\begin{array}{c} y_{1}\\ y_{2}\\ \vdots \\ y_{d} \end{array}\right)=P_{1}CP_{2}\left(\begin{array}{c} \lambda_{1}\\ \lambda_{2}\\ \vdots \\ \lambda_{d} \end{array}\right).$$
Putting $P_{1}CP_{2}=\left[w_{ij}\right]$ yields that
(10)$$\lambda_{i}=\sum_{j=1}^{d}w_{ij}\lambda_{j}\left(i=1,2,\ldots ,d\right).$$
Thus, when $\lambda_{1}=\lambda_{2}=\ldots =\lambda_{r}>\lambda_{r+1}\geq \ldots \geq \lambda_{d}$, we see from Equation (Equation 10) that for 1≤i≤r,
$$\lambda_{i}=\left(\sum_{j=1}^{r}w_{ij}\right)\lambda_{i}+\sum_{j=r+1}^{d}w_{ij}\lambda_{j}$$
and so $\sum_{j=1}^{r}w_{ij}=1,w_{ij}=0\left(1\leq i\leq r,r<j\leq d\right)$. Using Equation (Equation 10) again yields that for 1+r≤i≤d,
$$\lambda_{i}=\left(\sum_{j=1}^{r}w_{ij}\right)\lambda_{1}+\sum_{j=r+1}^{d}w_{ij}\lambda_{j}$$
and so $\sum_{j=1}^{r}w_{ij}=0$, implying that $w_{ij}=0\left(r<i\leq d,1\leq j\leq r\right)$. Thus,
(11)$$P_{1}CP_{2}=\left(\begin{array}{cccc} D_{1} & 0 & \ldots & 0\\ 0 & D_{2} & \ldots & 0\\ \vdots & \vdots & \ddots & \vdots \\ 0 & 0 & \ldots & D_{k} \end{array}\right),$$
where *k* means the number of different eigenvalues $\mu_{1}>\mu_{2}>\ldots >\mu_{k}$ of ρ and $D_{i}$ is an $r_{i}\times r_{i}$-doubly stochastic matrix, and $r_{i}$ denotes the multiplicity of the *i*th eigenvalue $\mu_{i}$.

Conversely, suppose that there exist d×d permutation matrices $P_{1}$ and $P_{2}$ such that $P_{1}CP_{2}$ is of the form (Equation 11) where k>1. Since the matrix $P_{1}CP_{2}$ can be written as
$$P_{1}CP_{2}=\left(\begin{array}{cccc} |\langle e_{s_{1}}|f_{t_{1}}\rangle {|}^{2} & |\langle e_{s_{1}}|f_{t_{2}}\rangle {|}^{2} & \cdots & |\langle e_{s_{1}}|f_{t_{d}}\rangle {|}^{2}\\ |\langle e_{s_{2}}|f_{t_{1}}\rangle {|}^{2} & |\langle e_{s_{2}}|f_{t_{2}}\rangle {|}^{2} & \cdots & |\langle e_{s_{2}}|f_{t_{d}}\rangle {|}^{2}\\ \vdots & \vdots & \ddots & \vdots \\ |\langle e_{s_{d}}|f_{t_{1}}\rangle {|}^{2} & |\langle e_{s_{d}}|f_{t_{2}}\rangle {|}^{2} & \cdots & |\langle e_{s_{d}}|f_{t_{d}}\rangle {|}^{2} \end{array}\right),$$
where
$$\left(\begin{array}{c} s_{1}\\ s_{2}\\ \vdots \\ s_{d} \end{array}\right)=P_{1}\left(\begin{array}{c} 1\\ 2\\ \vdots \\ d \end{array}\right),\left(\begin{array}{c} t_{1}\\ t_{2}\\ \vdots \\ t_{d} \end{array}\right)=P_{2}\left(\begin{array}{c} 1\\ 2\\ \vdots \\ d \end{array}\right),$$
we see from condition (Equation 11) that
(12)$$\begin{array}{c} \langle e_{s_{i}}|f_{t_{j}}\rangle =0\left(\forall r_{1}<j\leq d,1\leq i\leq r_{1}\right),\;\langle e_{s_{i}}|f_{t_{j}}\rangle =0\left(\forall r_{1}<i\leq d,1\leq j\leq r_{1}\right). \end{array}$$
This implies that the subspaces generated by $\{|e_{s_{i}}\rangle {\}}_{i=1}^{r_{1}}$ and $\{|f_{t_{j}}\rangle {\}}_{j=1}^{r_{1}}$ are equal and so
$$\rho :=\frac{1}{r_{1}}\sum_{i=1}^{r_{1}}|e_{s_{i}}\rangle \langle e_{s_{i}}|=\frac{1}{r_{1}}\sum_{j=1}^{r_{1}}|f_{t_{j}}\rangle \langle f_{t_{j}}|,$$
Clearly, $\rho \in I\left(e\right)\cap I\left(f\right)\backslash \left\{\frac{1}{d}I\right\}$.

As a conclusion, we arrive at the following.

**Theorem** **2.**
*Let d≥2, $e=\{|e_{i}\rangle {\}}_{i=1}^{d}$ and $f=\{|f_{j}\rangle {\}}_{j=1}^{d}$ be two orthonormal bases for H and set $C=\left[|\langle e_{i}|f_{j}\rangle {|}^{2}\right]$. Then there exists a state $\rho \neq \frac{1}{d}I$ in I(e)∩I(f) if and only if there exist two d×d permutation matrices $P_{1}$ and $P_{2}$ such that the matrix $P_{1}CP_{2}$ is k×k block-diagonal for some k>1.*


**Example** **2.**
*Let $d>3,e=\{|e_{i}\rangle {\}}_{i=1}^{d}$ and $f=\{|f_{j}\rangle {\}}_{j=1}^{d}$ be two orthonormal bases for H such that*

$$|\langle f_{i}|e_{j}\rangle |=\frac{1}{\sqrt{2}}\left(i,j=1,2\right),|e_{i}\rangle =|f_{i}\rangle \left(i=3,4,\ldots ,d\right).$$

*Then*

$$C=\left[|\langle e_{i}|f_{j}\rangle {|}^{2}\right]=\left(\begin{array}{ccccc} 0.5 & 0.5 & 0 & \cdots & 0\\ 0.5 & 0.5 & 0 & \cdots & 0\\ 0 & 0 & 1 & \cdots & 0\\ \vdots & \vdots & \vdots & \ddots & \vdots \\ 0 & 0 & \cdots & 0 & 1 \end{array}\right).$$

*It follows from Theorem 2 that there exists a state ρ∈I(e)⋂I(f)\{I/d}; for example,*

$$\rho =\frac{1}{d-2}\sum_{i=3}^{d}|e_{i}\rangle \langle e_{i}|.$$



**Remark** **2.**
*From Theorem 2, we know that whether I(e)⋂I(f)\{I/d}≠∅ depends on the structure of the matrix C given by Equation (Equation 7). Since this, we call C the correlation matrix of the bases e and f and denote it by $C_{e,f}$. Clearly, it can be written as the Hardamard product of the transition matrix $T_{e,f}$ from e to f and its conjugate matrix $T_{e,f}^{*}$:*

$$C_{e,f}=T_{e,f}⊙T_{e,f}^{*},$$

*where*

(13)
$$T_{e,f}=\left(\begin{array}{cccc} \langle e_{1}|f_{1}\rangle & \langle e_{1}|f_{2}\rangle & \cdots & \langle e_{1}|f_{d}\rangle \\ \langle e_{2}|f_{1}\rangle & \langle e_{2}|f_{2}\rangle & \cdots & \langle e_{2}|f_{d}\rangle \\ \vdots & \vdots & \ddots & \vdots \\ \langle e_{d}|f_{1}\rangle & \langle e_{d}|f_{2}\rangle & \cdots & \langle e_{d}|f_{d}\rangle \end{array}\right).$$



Theorem 2 also tells us that when $\langle e_{i}|f_{j}\rangle \neq 0$ for all i,j, there do not exist permutation matrices $P_{1}$ and $P_{2}$ such that $P_{1}CP_{2}$ is r×r(2≤r≤d) block diagonal, so $I\left(e\right)\cap I\left(f\right)=\left\{I/d\right\}$. Especially, for a pair of mutually unbiased bases *e* and *f*, when ρ∈I(e) and $\rho \neq \frac{1}{d}I$, we have ρ∉I(f). Conversely, when ρ is a maximally coherent state w.r.t. *e*, a question is: whether ρ is also maximally coherent w.r.t. *f*. The follow example shows that the answer is negative.

**Example** **3.**
*Let e={|0〉,|1〉} and $f=\left\{|f_{0}\rangle ,|f_{1}\rangle \right\}$ be a pair of mutually unbiased bases for $H=C^{2}$ where*

$$|f_{0}\rangle =\frac{1}{\sqrt{2}}(|0\rangle +\left|1\rangle ),\right|f_{1}\rangle =\frac{1}{\sqrt{2}}(|0\rangle -|1\rangle ),$$

*choose*

$$\rho_{1}=\frac{1}{2}(|f_{0}\rangle \langle f_{0}\left|+\right|f_{0}\rangle \langle f_{1}\left|+\right|f_{1}\rangle \langle f_{0}\left|+\right|f_{1}\rangle \langle f_{1}|)=|0\rangle \langle 0|.$$

*Then $\rho_{1}$ is maximally coherent with respect to f but is incoherent w.r.t. e, while for the state*

$$\rho_{2}=\frac{1}{2}(|f_{0}\rangle \langle f_{0}\left|+i\right|f_{0}\rangle \langle f_{1}\left|-i\right|f_{1}\rangle \langle f_{0}\left|+\right|f_{1}\rangle \langle f_{1}|),$$

*we have*

$$C_{e,\ell_{1}}\left(\rho_{2}\right)=C_{P_{f},\ell_{1}}\left(\rho_{2}\right)=1.$$

*Therefore, $\rho_{2}$ is both maximally coherent w.r.t. e and f.*


The following theorem shows that there must exist a maximally coherent state w.r.t. any two bases for $C^{2}$.

**Theorem** **3.**
*Let $e=\{|e_{i}\rangle {\}}_{i=1}^{2}$ and $f=\{|f_{j}\rangle {\}}_{j=1}^{2}$ be two orthonormal bases for $C^{2}$. Then there exists a state $\rho \in D(C^{2})$ such that*

$$C_{e,\ell_{1}}\left(\rho \right)+C_{f,\ell_{1}}\left(\rho \right)=2.$$



**Proof.** First, we observe that $C_{e,\ell_{1}}\left(\rho \right)=1$ if and only if
(14)$$\rho =\frac{1}{2}(|e_{1}\rangle \langle e_{1}|+e^{i\alpha }|e_{1}\rangle \langle e_{2}|+e^{-i\alpha }|e_{1}\rangle \langle e_{2}\left|+\right|e_{2}\rangle \langle e_{2}|)$$
and $C_{P_{f},\ell_{1}}\left(\rho \right)=1$ if and only if
(15)$$\rho =\frac{1}{2}(|f_{1}\rangle \langle f_{1}|+e^{i\beta }|f_{1}\rangle \langle f_{2}|+e^{-i\beta }|f_{2}\rangle \langle f_{1}\left|+\right|f_{2}\rangle \langle f_{2}|).$$
Suppose that
$$|f_{1}\rangle =u_{11}|e_{1}\rangle +u_{12}|e_{2}\rangle ,|f_{2}\rangle =u_{21}|e_{1}\rangle +u_{22}|e_{2}\rangle ,$$
then $U:=[u_{ij}]$ is a unitary matrix, which is given.For a state ρ of the form given by (Equation 14), then $C_{e,\ell_{1}}\left(\rho \right)=1$. We compute that
〈f1|ρ|f1〉=(u11*〈e1|+u12*〈e2|)|ρ|(u11|e1〉+u12|e2〉)=12(|u11|2+u11*u12eiα+u11u12*e−iα+|u12|2)=12+Re(u11*u12eiα),
$$\begin{array}{ccc} \langle f_{1}|\rho |f_{2}\rangle & = & (u_{11}^{*}\langle e_{1}|+u_{12}^{*}\langle e_{2}|)\left|\rho \right|(u_{21}|e_{1}\rangle +u_{22}|e_{2}\rangle )\\ & = & \frac{1}{2}\left(u_{11}^{*}u_{21}+u_{11}^{*}u_{22}e^{i\alpha }+u_{12}^{*}u_{21}e^{-i\alpha }+u_{12}^{*}u_{22}\right)\\ & = & \frac{1}{2}\left(u_{11}^{*}u_{22}e^{i\alpha }+u_{12}^{*}u_{21}e^{-i\alpha }\right), \end{array}$$
〈f2|ρ|f2〉=(u21*〈e1|+u22*〈e2|)|ρ|(u21|e1〉+u22|e2〉)=12(|u21|2+u21*u22eiα+u21u22*e−iα+|u22|2)=12+Re(u21*u22eiα).
Thus, $C_{f,\ell_{1}}\left(\rho \right)=1$ if and only if
(16)Re(u11*u12eiα)=0;u11*u22eiα+u12*u21e−iα=eiβ;Re(u21*u22eiα)=0,
if and only if
(17)Re(u11*u12eiα)=0;u11*u22eiα+u12*u21e−iα=eiβ
since $u_{11}^{*}u_{12}=-u_{21}^{*}u_{22}$.Since *U* is a unitary matrix, it can be represented as
$$U=\left(\begin{array}{cc} u_{11} & u_{12}\\ u_{21} & u_{22} \end{array}\right)=\left(\begin{array}{cc} re^{i\theta_{1}} & \sqrt{1-r^{2}}e^{i\theta_{2}}\\ \sqrt{1-r^{2}}e^{i\theta_{3}} & re^{i\theta_{4}} \end{array}\right)$$
where 0≤r≤1, and $\theta_{k}\in R$ s.t. $e^{i(\theta_{1}-\theta_{3})}+e^{i(\theta_{2}-\theta_{4})}=0$. The last condition implies that $-\theta_{1}+\theta_{2}+\theta_{3}-\theta_{4}=\left(2n+1\right)\pi$ for some integer *n*. Taking $\alpha =(\theta_{1}-\theta_{2}+\theta_{3}-\theta_{4})/2$ implies that $|u_{11}^{*}u_{22}e^{i\alpha }+u_{12}^{*}u_{21}e^{-i\alpha }|=1$ and so there exists a real number β such that second equation in (Equation 17) holds. Since $-\theta_{1}+\theta_{2}+\alpha =n\pi +\pi /2$, the first equation in (Equation 17) holds too. Hence, $C_{f,\ell_{1}}\left(\rho \right)=1.$This shows that the state ρ defined by Equation (Equation 14) with $\alpha =(\theta_{1}-\theta_{2}+\theta_{3}-\theta_{4})/2$ satisfies
$$C_{e,\ell_{1}}\left(\rho \right)=C_{f,\ell_{1}}\left(\rho \right)=1,$$
that is, $C_{e,\ell_{1}}\left(\rho \right)+C_{f,\ell_{1}}\left(\rho \right)=2.$ □

## 3. Weak Coherence

In this section, we turn to discuss the weak coherence of quantum states. To this, we use B to denote a set of *k* orthonormal bases $e^{1},e^{2},\ldots ,e^{k}$ for *H*, i.e., $B=\{e^{1},e^{2},\ldots ,e^{k}\}.$

**Definition** **1.**
*We say that ρ∈D(H) is strongly incoherent (S-incoherent) w.r.t. B if ρ is incoherent w.r.t. any basis in B. Otherwise, we say that ρ is weakly coherent (W-coherent) w.r.t. B.*


Denoted by SI(B) the set of all S-incoherent states of *H* w.r.t. B. Clearly,
$$\frac{1}{d}I\in \mathrm{SI}\left(B\right)=\overset{k}{\underset{i=1}{⋂}}I\left(e^{i}\right).$$

**Definition** **2.**
*Let *Φ* be a quantum operation on B(H). Then *Φ* is said to be an S-incoherent operation (SIO) w.r.t. B (or B-incoherent operation (BIO)) if $\Phi \in \mathrm{IO}(e^{i})$ for all i=1,2,…,k, that is, for each i=1,2,…,k, *Φ* has a family of Kraus operators ${\left\{E_{in}\right\}}_{n=1}^{m_{i}}$ such that*

$$E_{in}\left(I\left(e^{i}\right)\right)E_{in}^{†}\subset R^{+}I\left(e^{i}\right),\;\;\forall n=1,2,\ldots ,m_{i}.$$



Denoted by IO(B) the set of all SIOs w.r.t. B, then
$$\mathrm{IO}\left(B\right)=\overset{k}{\underset{i=1}{⋂}}\mathrm{OI}\left(e^{i}\right).$$

Similar to the definition of the standard coherence measure, let us introduce the concept of a B-coherence measure.

**Definition** **3.**
*A function $C_{B}:D\left(H\right)\to R$ is said to be a B-coherence measure if the following four conditions are satisfied:*

*(1) Faithfulness: $\forall \rho \in D\left(H\right),C_{B}\left(\rho \right)\geq 0$; $C_{B}\left(\rho \right)=0$ if and only if ρ∈SI(B).*

*(2) Monotonicity: $C_{B}\left(\Phi \left(\rho \right)\right)\leq C_{B}\left(\rho \right)$ for every Φ∈IO(B) and for every ρ∈D(H).*

*(3) Strong monotonicity: for each i=1,2,…,k,$\sum_{n=1}^{m_{i}}p_{in}C_{B}\left(\rho_{in}\right)\leq C_{B}\left(\rho \right)$ for every ρ∈D(H) and every Φ∈IO(B) with a family Kraus operators ${\left\{E_{in}\right\}}_{n=1}^{m_{i}},$ where $p_{in}=\mathrm{tr}\left(E_{in}\rho E_{in}^{†}\right)$ and $\rho_{in}=\frac{1}{p_{in}}E_{in}\rho E_{in}^{†}$ for $p_{in}>0$, and $\rho_{in}=\frac{1}{d}I$ for $p_{in}=0$.*

*(4) Convexity: $C_{B}\left(\sum_{n=1}^{m}p_{n}\rho_{n}\right)\leq \sum_{n=1}^{m}p_{n}C_{B}\left(\rho_{n}\right),$ where $\rho_{n}\in D\left(H\right)\left(n=1,2,\ldots ,m\right)$ and ${\left\{p_{n}\right\}}_{n=1}^{m}$ is a probability distribution.*


The following theorem gives a method for constructing a B-coherence measure from *k*$e^{i}$-coherence measures (i=1,2,…,k).

**Theorem** **4.**
*Let $C_{e^{i}}\left(i=1,2,\ldots ,k\right)$ be $e^{i}$-coherence measures. Then the function $C_{B}:D\left(H\right)\to R$ defined by*

(18)
$$C_{B}\left(\rho \right)=\sum_{i=1}^{k}C_{e^{i}}\left(\rho \right)\left(\forall \rho \in D\left(H\right)\right)$$

*is a B-coherence measure.*


**Proof.** (1) Let ρ∈D(H). Since $C_{e^{i}}\left(\rho \right)\geq 0$ for all $e^{i}\left(i=1,2,\ldots ,k\right),$ we have $C_{B}\left(\rho \right)=\sum_{i=1}^{k}C_{e^{i}}\left(\rho \right)\geq 0.$ Furthermore,
$$\sum_{i=1}^{k}C_{e^{i}}\left(\rho \right)=0\Leftrightarrow C_{e^{i}}\left(\rho \right)=0\left(i=1,2,\ldots ,k\right)\Leftrightarrow \rho \in \mathrm{SI}\left(B\right).$$(2) Let Φ∈IO(B). For each i=1,2…,k, since $C_{e^{i}}$ is an $e^{i}$-coherence measure and $\Phi \in \mathrm{IO}(e^{i})$, we get
$$C_{e^{i}}\left(\Phi \left(\rho \right)\right)\leq C_{e^{i}}\left(\rho \right)$$
for all ρ∈D(H), and so
$$C_{B}\left(\Phi \left(\rho \right)\right)=\sum_{i=1}^{k}C_{e^{i}}\left(\Phi \left(\rho \right)\right)\leq \sum_{i=1}^{k}C_{e^{i}}\left(\rho \right)=C_{B}\left(\rho \right).$$(3) Let ρ∈D(H), Φ∈IO(B) with families of Kraus operators ${\left\{E_{in}\right\}}_{n=1}^{m_{i}}\left(i=1,2,\ldots ,k\right)$. Put $p_{in}=\mathrm{tr}\left(E_{in}\rho E_{in}^{†}\right)$ and $\rho_{in}=\frac{1}{p_{in}}E_{in}\rho E_{in}^{†}$ for $p_{in}>0$, and $\rho_{in}=\frac{1}{d}I$ for $p_{in}=0$. For each j=1,2,…,k, since $C_{e^{j}}$ is an $e^{j}$-coherence measure and $\Phi \in \mathrm{IO}(e^{j})$, we get
$$\sum_{n=1}^{m_{i}}p_{in}C_{e^{j}}\left(\rho_{in}\right)\leq C_{e^{j}}\left(\rho \right)\left(i,j=1,2,\ldots ,k\right).$$
This implies that for each i=1,2,…,k,
$$\sum_{n=1}^{m_{i}}p_{in}C_{B}\left(\rho_{in}\right)=\sum_{n=1}^{m_{i}}p_{in}\left(\sum_{j=1}^{k}C_{e^{j}}\left(\rho_{in}\right)\right)=\sum_{j=1}^{k}\left(\sum_{n=1}^{m_{i}}p_{in}C_{e^{j}}\left(\rho_{in}\right)\right)\leq \sum_{j=1}^{k}C_{e^{j}}\left(\rho \right)=C_{B}\left(\rho \right).$$(4) Let $\rho_{n}\in D\left(H\right)\left(n=1,2,\ldots ,m\right)$ and let ${\left\{p_{n}\right\}}_{n=1}^{m}$ be a probability distribution. Since $C_{e^{i}}$ is an $e^{i}$-coherence measure, we have
$$\sum_{n=1}^{m}p_{n}C_{e^{i}}\left(\rho_{n}\right)\geq C_{e^{i}}\left(\sum_{n=1}^{m}p_{n}\rho_{n}\right)$$
for all i=1,2,…,k, and therefore,
$$\sum_{n=1}^{m}p_{n}C_{B}\left(\rho_{n}\right)=\sum_{i=1}^{k}\left(\sum_{n=1}^{m}p_{n}C_{e^{i}}\left(\rho_{n}\right)\right)\geq \sum_{i=1}^{k}C_{e^{i}}\left(\sum_{n=1}^{m}p_{n}\rho_{n}\right)=C_{B}\left(\sum_{n=1}^{m}p_{n}\rho_{n}\right).$$
Using Definition 3 yields that the function $C_{B}$ defined by Equation (Equation 18) becomes a B-coherence measure. □

Using Theorem 4 yields that the function $C_{B}:D\left(H\right)\to R$ defined by
(19)$$C_{B,\ell_{1}}\left(\rho \right)=\sum_{i=1}^{k}C_{e^{i},\ell_{1}}\left(\rho \right)\left(\forall \rho \in D\left(H\right)\right)$$
is a B-coherence measure. We see from property (Equation 1) that $C_{B,\ell_{1}}\left(\rho \right)\leq k\left(d-1\right)$ for all states ρ of the system. A state ρ is said to be *maximally coherent* w.r.t. $C_{B,\ell_{1}}$ if $C_{B,\ell_{1}}\left(\rho \right)=k\left(d-1\right)$. Clearly, a state ρ is maximally coherent $C_{B,\ell_{1}}$ if and only if it is maximally coherent w.r.t. each $C_{e^{i},\ell_{1}}$.


**Remark 3.**
*(1) $\frac{I}{d}\in \mathrm{SI}\left(B\right)$; Especially, if there exist two mutually unbiased bases in B, then $\mathrm{SI}\left(B\right)=\left\{\frac{I}{d}\right\}$, that is, $C_{B,\ell_{1}}\left(\rho \right)=0$ if and only if $\rho =\frac{I}{d}$.*

*(2) Theorem 3 implies when d=2 and B={e,f}(e≠f), there exists a maximally coherent state ρ w.r.t. $C_{B,\ell_{1}}$, that is, $C_{B,\ell_{1}}\left(\rho \right)=2$.*

*(3) The following theorem means that when d=2 and B={e,f,g} is a complete set of mutually unbiased bases, there does not exist necessarily a maximally coherent state w.r.t. $C_{B,\ell_{1}}$.*


It was proved in [47] that the maximal number MUB(H) of mutually unbiased bases for *H* is d+1 if the dimension *d* of *H* is a prime-power. Thus, $MUB(C^{2})=3$, i.e., there exists a complete set of three mutually unbiased bases for $C^{2}$.

**Theorem** **5.**
*Let B={e,f,g} where $e=\{|e_{1}\rangle ,|e_{2}\rangle \}$ be any orthonormal basis for $C^{2}$, $f=\{|f_{1}\rangle ,|f_{2}\rangle \}$ and $g=\{|g_{1}\rangle ,|g_{2}\rangle \}$ with*

$$|f_{1}\rangle =\frac{1}{\sqrt{2}}(|e_{1}\rangle +|e_{2}\rangle ),|f_{2}\rangle =\frac{1}{\sqrt{2}}(|e_{1}\rangle -|e_{2}\rangle ),$$


$$|g_{1}\rangle =\frac{1}{\sqrt{2}}(|e_{1}\rangle +i|e_{2}\rangle ),|f_{2}\rangle =\frac{1}{\sqrt{2}}(|e_{1}\rangle -i|e_{2}\rangle ).$$

*Then e,f and g are mutually unbiased bases pairwise for $C^{2}$ and $C_{B,\ell_{1}}\left(\rho \right)<3$ for all states ρ of $C^{2}$, that is, there does not exist a state ρ such that*

(20)
$$C_{e,\ell_{1}}\left(\rho \right)=C_{f,\ell_{1}}\left(\rho \right)=C_{g,\ell_{1}}\left(\rho \right)=1.$$



**Proof.** Obviously, e,f and *g* are mutually unbiased bases pairwise for $C^{2}$. Suppose that there exists a state ρ such that Equation (Equation 20) holds, i.e.,
(21)$$|\langle e_{1}\left|\rho \right|e_{2}\rangle |=|\langle f_{1}\left|\rho \right|f_{2}\rangle |=|\langle g_{1}\left|\rho \right|g_{2}\rangle |=\frac{1}{2}.$$
Then under the three bases, we have
(22)$$\rho =a|e_{1}\rangle \langle e_{1}|+\frac{1}{2}e^{i\theta_{1}}|e_{1}\rangle \langle e_{2}|+\frac{1}{2}e^{-i\theta_{1}}|e_{2}\rangle \langle e_{1}\left|+\left(1-a\right)\right|e_{2}\rangle \langle e_{2}|,$$
(23)$$\rho =b|f_{1}\rangle \langle f_{1}|+\frac{1}{2}e^{i\theta_{2}}|f_{1}\rangle \langle f_{2}|+\frac{1}{2}e^{-i\theta_{2}}|f_{2}\rangle \langle f_{1}\left|+\left(1-b\right)\right|f_{2}\rangle \langle f_{2}|,$$
(24)$$\rho =c|g_{1}\rangle \langle g_{1}|+\frac{1}{2}e^{i\theta_{3}}|g_{1}\rangle \langle g_{2}|+\frac{1}{2}e^{-i\theta_{3}}|g_{2}\rangle \langle g_{1}\left|+\left(1-c\right)\right|g_{2}\rangle \langle g_{2}|,$$
where $a,b,c\in \left[0,1\right],0\leq \theta_{k}<2\pi \left(k=1,2,3\right).$ Since ρ≥0, we conclude from Equation (Equation 21) that $a=b=c=\frac{1}{2}$. Substituting $2|f_{i}\rangle \langle f_{j}|$ in Equation (Equation 23) with
$$2|f_{1}\rangle \langle f_{1}\left|=\right|e_{1}\rangle \langle e_{1}\left|+\right|e_{1}\rangle \langle e_{2}\left|+\right|e_{2}\rangle \langle e_{1}\left|+\right|e_{2}\rangle \langle e_{2}|,$$
$$2|f_{1}\rangle \langle f_{2}\left|=\right|e_{1}\rangle \langle e_{1}\left|-\right|e_{1}\rangle \langle e_{2}\left|+\right|e_{2}\rangle \langle e_{1}\left|-\right|e_{2}\rangle \langle e_{2}|,$$
$$2|f_{2}\rangle \langle f_{1}\left|=\right|e_{1}\rangle \langle e_{1}\left|+\right|e_{1}\rangle \langle e_{2}\left|-\right|e_{2}\rangle \langle e_{1}\left|-\right|e_{2}\rangle \langle e_{2}|,$$
$$2|f_{2}\rangle \langle f_{2}\left|=\right|e_{1}\rangle \langle e_{1}\left|-\right|e_{1}\rangle \langle e_{2}\left|-\right|e_{2}\rangle \langle e_{1}\left|+\right|e_{2}\rangle \langle e_{2}|,$$
and comparing the coefficient of $|e_{1}\rangle \langle e_{2}|$ in Equations (Equation 22) and (Equation 23), we find that
(25)$$\begin{array}{c} e^{i\theta_{1}}=-i\sin\theta_{2}\;\mathrm{and}\;\mathrm{so}\;\cos\theta_{1}=0. \end{array}$$
Similarly, substituting $2|g_{i}\rangle \langle g_{j}|$ in Equation (Equation 24) with
$$2|g_{1}\rangle \langle g_{1}\left|=\right|e_{1}\rangle \langle e_{1}\left|-i\right|e_{1}\rangle \langle e_{2}\left|+\right|e_{2}\rangle \langle e_{1}\left|+i\right|e_{2}\rangle \langle e_{2}|,$$
$$2|g_{1}\rangle \langle g_{2}\left|=\right|e_{1}\rangle \langle e_{1}\left|+i\right|e_{1}\rangle \langle e_{2}\left|+i\right|e_{2}\rangle \langle e_{1}\left|-\right|e_{2}\rangle \langle e_{2}|,$$
$$2|g_{2}\rangle \langle g_{1}\left|=\right|e_{1}\rangle \langle e_{1}\left|-i\right|e_{1}\rangle \langle e_{2}\left|-i\right|e_{2}\rangle \langle e_{1}\left|-\right|e_{2}\rangle \langle e_{2}|,$$
$$2|g_{2}\rangle \langle g_{2}\left|=\right|e_{1}\rangle \langle e_{1}\left|+i\right|e_{1}\rangle \langle e_{2}\left|-i\right|e_{2}\rangle \langle e_{1}\left|+\right|e_{2}\rangle \langle e_{2}|,$$
and comparing the coefficient of $|e_{1}\rangle \langle e_{2}|$ in Equations (Equation 22) and (Equation 24), we find that
(26)$$\begin{array}{c} e^{i\theta_{1}}=-\sin\theta_{3}\;\mathrm{and}\;\mathrm{so}\;\sin\theta_{1}=0. \end{array}$$
Combining Equations (Equation 25) and (Equation 26) yields that $\cos\theta_{1}=\sin\theta_{1}=0$, a contradiction. □

## 4. Conclusions

In this paper, we have introduced a correlation function m(e,f) of two orthonormal bases *e* and *f* with the property that $0\leq m\left(e,f\right)\leq d^{\frac{3}{2}}-d$, and proved that m(e,f)=0 if and only if the rank-one projective measurements generated by *e* and *f* are identical if and only if I(e)=I(f), where I(e) and I(f) denote the sets of incoherent states with respect to *e* and *f*, respectively. We have also shown that m(e,f) reaches the maximum $d^{\frac{3}{2}}-d$ if and only if the bases *e* and *f* are mutually unbiased; in that case, the intersection I(e)⋂I(f) includes only the maximally mixed state. We have observed that even though two bases *e* and *f* are not mutually unbiased, I(e)⋂I(f) may include only the maximally mixed state. We have obtained a necessary and sufficient condition for $I\left(e\right)⋂I\left(f\right)=\frac{I}{d}$. We have introduced the concepts of strong incoherence and weak coherence of a quantum state w.r.t. a set B of *k* orthonormal bases and proposed a measure $C_{B}$ for the weak coherence. In the two-qubit system, we have proved that there exists a maximally coherent state w.r.t. the measure $C_{B,\ell_{1}}$ when B consists of any two bases and observed that there exist does not a maximally coherent state w.r.t. the measure $C_{B,\ell_{1}}$ when B consists of some three mutually unbiased bases.

## Figures and Tables

**Figure 1 entropy-24-00659-f001:**
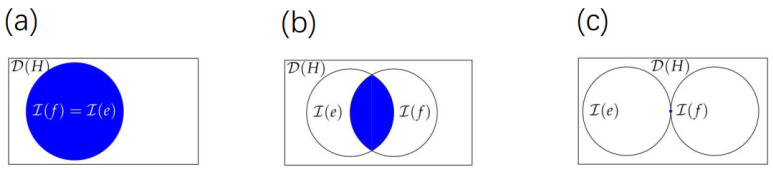
Relationships between m(e,f) and I(e)⋂I(f), where subfigures (**a**–**c**) correspond to the cases that m(e,f)=0, m(e,f)>0 and $m\left(e,f\right)=d^{\frac{3}{2}}-d$, respectively.

## Data Availability

Data are contained within the article.

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
