# Peer review of "Quantum Incoherence Based Simultaneously on k Bases"

_entropy, 2022, doi:10.3390/e24050659_

Round 1
Author Response
Dear the reviewer,
Thank you for your kind comments and useful suggestions. Our manuscript has been revised accordingly. The following points are our responses about your comments.
- In the introduction, the authors stated the area of focus in one sentence. How the investigations/results presented in this paper will affect the aforementioned fields directly or indirectly is not mentioned. The introduction should include that.
Response: The title was revised and some words were added in the past part of the introduction. The changed and revised places are red-colored.
- In line 51, ρ is missing from the sentence “...for all 2 D(H)...”.
Response: added.
- The last equation of page 3, the expression of ρ containing µi is written as ρ =\frac{1}{d}\sum_{j}|f_j><f_j| on the next page. This is not clear to me.
Response: revised.
- The paper PRL 126, 220404 (2021) defines the term “Set-coherence” in a different sense. The authors of the current paper should make comment about it to avoid confusion for the readers.
Response: The “set-coherence” in [PRL 126, 220404 (2021)] is cited and reviewed in the introduction, and the concept of “set-coherence” in our paper has been deleted and changed as “strong-incoherence” and “weak” in Definition 1. The definitions of incoherent operation and coherence measure have been revised accordingly in Definition 2 and Definition 3, respectively. Based on these new definitions, Theorem 4 have been improved and proved.
Thank you again. Some further comments are welcome.
Reviewer 2 Report
I cannot recommend in favor of publication of this manuscript in its present form.
Most of the results presented are simple (in some cases trivial) results in linear algebra which cannot be published as such. However, some of them may have an interest in applications to quantum theory (in particular in quantum computation, information, and communication).
I suggest that the paper is written in a much terser form for the mathematical part, while the main emphasis is put on new perspective applications.
Author Response
DearDear the reviewer,
Dear the reviewer,
Thank you for your comments. The following points are our responses about your comments and another reviewer.
- The title was revised and some words were added in the past part of the introduction. The changed and revised places are red-colored.
- The last equation of page 3, ρ =\frac{1}{d}\sum_{j}|f_j><f_j| is reproved.
- The “set-coherence” in [PRL 126, 220404 (2021)] is cited and reviewed in the introduction, and the concept of “set-coherence” in our paper has been deleted and changed as “strong-incoherence” and “weak” in Definition 1. The definitions of incoherent operation and coherence measure have been revised accordingly in Definition 2 and Definition 3, respectively. Based on these new definitions, Theorem 4 have been improved and proved.
Thank you again. Some further comments are welcome.
Round 2
Reviewer 1 Report
The authors have clarified the queries regarding the paper. Considering the approval from other referee(s), it is suggested that the paper be accepted in the present form.
Author Response
The methods is adequately described as "using Matrix Theory Method" in the abstract.
Reviewer 2 Report
The authors have amended the manuscript complying with the first report's comments and recommendations. In my opinion, the manuscript is now woth publication.
Author Response
Some places have been revised (red-colored) and 9 references have been added and cited.